# A Novel Synthetic Oleanolic Acid Derivative Inhibits Glioma Cell Proliferation by Regulating Cell Cycle G2/M Arrest

**DOI:** 10.3390/ph16050642

**Published:** 2023-04-24

**Authors:** Tai-Hsin Tsai, Cheng-Yu Tsai, Sin-Hua Moi, Chieh-Hsin Wu, Kuan-Ting Lee, Yi-Chiang Hsu, Yu-Feng Su

**Affiliations:** 1Division of Neurosurgery, Department of Surgery, Kaohsiung Medical University Hospital, Kaohsiung 80756, Taiwan; tsaitaihsin@gmail.com (T.-H.T.); moutzyy691010@yahoo.com.tw (C.-Y.T.); moi9009@gmail.com (S.-H.M.); wujoeys@gmail.com (C.-H.W.); 2Department of Surgery, School of Medicine, College of Medicine, Kaohsiung Medical University, Kaohsiung 80756, Taiwan; 3Graduate Institute of Medicine, College of Medicine, Kaohsiung Medical University, Kaohsiung 80756, Taiwan; ayta860404@gmail.com; 4Division of Neurosurgery, Department of Surgery, Kaohsiung Municipal Ta-Tung Hospital, Kaohsiung 80145, Taiwan; 5School of Medicine, I-Shou University, Kaohsiung 82445, Taiwan

**Keywords:** RTA-dh404, CDDO-dhTFEA, glioblastoma, cell cycle arrest

## Abstract

2-Cyano-3,12-dioxooleana-1,9(11)-dien-28-oic acid-9,11-dihydro-trifluoroethyl amide (CDDO-dhTFEA) has antioxidant and anti-inflammatory activities; however, whether CDDO-dhTFEA has anticancer effects is unclear. The objective of this research was to investigate the possibility of CDDO-dhTFEA as a potential cancer-fighting treatment in glioblastoma cells. Our experiments were performed on U87MG and GBM8401 cells, and we found that CDDO-dhTFEA was effective in reducing cell proliferation in both cell lines, in a manner that was dependent on both time and concentration. Additionally, we observed that CDDO-dhTFEA had a significant impact on the regulation of cell proliferation, which was evident in the increase in DNA synthesis that was observed in both cell types. CDDO-dhTFEA induced G2/M cell cycle arrest and mitotic delay, which may be associated with the inhibition of proliferation. Treatment with CDDO-dhTFEA led to cell cycle G2/M arrest and inhibited proliferation of U87MG and GBM8401 cells by regulating G2/M cell cycle proteins and gene expression in GBM cells in vitro.

## 1. Introduction

The invasiveness and aggressive biological behaviors of glioblastoma, the most aggressive primary brain tumor, are well-known factors that lead to high morbidity and mortality rates [1]. The current standard treatment for newly diagnosed glioblastoma is a combination of surgery, radiotherapy, and temozolomide chemotherapy [2,3]. However, this approach often fails to prevent recurrence, and there is currently no established standard treatment for recurrent glioblastoma [4]. As a result, there is a critical need to gain a comprehensive understanding of the molecular mechanisms of glioblastoma to facilitate the development of effective anticancer drugs.

Oleanolic acid (OA) is a natural pentacyclic triterpenoid with mild biological activity; OA also as a scaffold has potential for the creation of semi-synthetic triterpenoids with enhanced biological properties [5,6,7]. However, one of the challenges of using oleanolic acid or its derivatives for the treatment of brain tumors is the ability of the compound to cross the blood-brain barrier (BBB). Studies have shown that oleanolic acid has low bioavailability and poor BBB penetration due to its high molecular weight and low lipophilicity [8,9]. However, recent research has explored different strategies to improve the BBB permeability of oleanolic acid, including the use of nanoparticles, drug delivery systems, and chemical modifications [5].These approaches may offer new opportunities for the development of oleanolic acid-based therapies for brain diseases.

OA-derived synthetic triterpenoids, as shown in Figure 1, which have undergone a series of chemical modifications, exhibit increased solubility, bioavailability, and potency compared to the natural compound [7]. Among these synthetic compounds, 2-cyano-3,12-dioxoolean-1,9-dien-28-oic acid (CDDO) is a chemically modified compound with pleiotropic effects [10]. CDDO exhibits anti-inflammatory and antioxidative stress effects at low doses, induces cell differentiation at intermediate doses, and exerts cytotoxic effects at high doses [10]. Oleanolic acid-derived synthetic triterpenoids, such as CDDO, show great promise in the prevention and treatment of cancer due to their pleiotropic effects. Another oleanolic acid-derived synthetic triterpenoid, CDDO-dhTFEA, activates the Nrf2-Keap1 pathway [11,12], inhibits the NF-κB inflammatory pathway, and detoxifies reactive oxygen species [13].

Eukaryotic cells rely on tight regulation of the cell cycle to ensure consistent cell growth, and loss of control over the cell cycle can result in the development of cancer cells that continue to divide uncontrollably [14]. As a result, inducing cell cycle arrest and promoting cell apoptosis or autophagy are important therapeutic strategies for treating cancer cells with anticancer drugs [15,16,17]. CDDO and its derivatives have shown promise in inducing cell cycle arrest in a variety of cancers, such as breast cancer [18], leukemia [19], neuroblastoma [20,21], and glioma [22,23]. CDDO-dhTFEA has been shown to regulate the cancer cell cycle and induce cell cycle arrest. The administration of CDDO-dhTFEA resulted in a substantial rise in the amount of cells in the G2/M phase. Furthermore, the mitotic index assay revealed that CDDO-dhTFEA induced mitotic arrest in these cells in a dose-dependent manner. These findings are significant, as they suggest that CDDO-dhTFEA’s impact at a fundamental level suggests its potential as a promising candidate for chemotherapy treatment.

The aim of this study was to investigate the anticancer effects of CDDO-dhTFEA in glioblastoma cells, as well as the underlying mechanisms. The results from our experiments suggest that CDDO-dhTFEA treatment led to a significant suppression of cell proliferation in both U87MG and GBM8401 cells, and coincided with an increase in G2/M cell cycle arrest. The findings of this study indicate that CDDO-dhTFEA could be a promising candidate for glioblastoma therapy.

## 2. Results

### 2.1. CDDO-dhTFEA Exerts Time- and Dose-Dependent Inhibition of Cell Viability in Glioblastoma Cells

To assess the selectivity of CDDO-dhTFEA as an anticancer agent, additional experiments were performed. Specifically, U87MG and GBM8401 cell lines were treated with varying concentrations of CDDO-dhTFEA (ranging from 1 to 8 µM) for different time periods of 24, 48, and 72 h. Subsequently, the cell viability was assessed using the PrestoBlue reagent assay. Consistent with previous results, the data indicated that CDDO-dhTFEA exhibited a significant inhibitory effect on the proliferation of both cells (Figure 2). Furthermore, the data indicated that treatment with CDDO-dhTFEA, at varying concentrations ranging from 1 to 8 µM, resulted in a reduction in cell viability that was dose dependent, and the decrease in cell viability at different time points (24, 48, and 72 h) was time dependent, further supporting its ability to inhibit clone formation in these cells in a dose- and time-dependent manner. There was no change in normal lung fibroblasts MRC-5 cells (data not shown). These results suggest that CDDO-dhTFEA may have the potential to become a potent and specific antitumor agent with activity against glioblastoma.

### 2.2. CDDO-dhTFEA Regulates Proliferation and DNA Synthesis in Glioblastoma Cells

We investigated the effects of CDDO-dhTFEA on glioma cell proliferation and DNA synthesis (Figure 3). Treatment with CDDO-dhTFEA at varying concentrations (from 0 to 8 µM) for 24 h was performed, and the EdU assay was utilized to measure cell proliferation and DNA synthesis. Interestingly, the results showed that the impact of CDDO-dhTFEA on DNA synthesis differed between the two cell lines. Specifically, the treatment significantly increased DNA synthesis in GBM8401 cells at 8 µM (*p* < 0.05) and in U87MG cells at 2 µM (*p* < 0.05). Interestingly, the drug response of the two cell lines differed with increasing concentrations. DNA synthesis increased linearly in the GBM8401 cells and decreased linearly in the U87MG cells with increasing drug concentration. Since the malignant degree of these two cell lines is different, we observed different results in the EdU assay. The cell cycle arrest was more apparent in GBM8401, and we obtained similar results in gene and protein expression levels. These results suggest that CDDO-dhTFEA modulates cell proliferation by regulating DNA synthesis during the cell cycle. Further investigation into the mechanism by which CDDO-dhTFEA regulates DNA synthesis revealed that it may affect the expression of key cell cycle regulators. In addition, a time course experiment revealed that the effect of CDDO-dhTFEA on DNA synthesis was time dependent, with the greatest increase observed after 24 h of treatment.

### 2.3. CDDO-dhTFEA Promotes the Accumulation of Cells in the G2/M Phase of the Cell Cycle

To evaluate the potential of CDDO-dhTFEA to induce cell cycle arrest in glioma cells, the experimental cells were subjected to varying concentrations ranging from 0 to 8 µM for a certain period, and then PI staining was used to analyze the cell cycle distribution (Figure 4). The glioma cells showed an increase in the proportion of cells in the G2/M phase following treatment with CDDO-dhTFEA. This suggests that CDDO-dhTFEA has the ability to reduce the number of cells entering the mitotic phase, ultimately leading to cell cycle arrest. The increase in the number of cells in G1 phase in both cell lines at different concentrations of CDDO-dhTFEA indicates that the compound may affect cell cycle progression differently in each cell line. These findings support the hypothesis that CDDO-dhTFEA inhibits the proliferation of glioblastoma cells and may serve as a potential therapeutic agent for glioblastoma treatment.

### 2.4. CDDO-dhTFEA Causes Mitotic Arrest in Gioblastoma Cells

Further investigation into the mechanism of cell cycle arrest involved the use of MPM-2 staining. As depicted in Figure 5, treatment with CDDO-dhTFEA caused a statistically significant increase in MPM-2 staining in both U87MG and GBM8401 cells, indicating the induction of mitotic arrest. These results suggest that treatment with CDDO-dhTFEA promotes protein synthesis during mitosis in U87MG and GBM8401 cells, which leads to mitotic arrest. Additionally, the potential for CDDO-dhTFEA to induce cell cycle arrest was further demonstrated by the increase in the number of G2/M cells in both cell lines following treatment with CDDO-dhTFEA, as shown in Figure 4. Taken together, these results suggest that CDDO-dhTFEA promotes DNA synthesis, leading cells to progress from the S phase to the G2 phase, where it induces mitotic arrest.

### 2.5. CDDO-dhTFEA Impacts the Relative Intensities of Closely Associated Cell Cycle Proteins in Glioma Cells

A Western blot analysis was employed to elucidate the mechanism behind the suppression of cellular growth and the induction of cell cycle arrest (Figure 6). The expression levels of various proteins involved in the regulation of the G2/M cell cycle were measured, including CyclinB1, CDK1, p21, WEE1, CDC25C, GADD45α, GADD45β, and GADD45γ. The results indicated that CDDO-dhTFEA treatment significantly downregulated the expression levels of CyclinB1, CDK1, GADD45α, GADD45β, and GADD45γ in GBM8401 cells. Meanwhile, the expression levels of Wee1 and CDC25C was upregulated in both U87MG and GBM8401 cells. Moreover, the CDDO-dhTFEA treatment led to an increase in the expression level of the cyclin-dependent kinase 1 inhibitor, p21, in GBM8401 cells. These proteins are known to play critical roles in the regulation of the G2/M cell cycle. These findings suggest that CDDO-dhTFEA can modulate the expression of cell cycle-associated proteins, including p21, WEE1, CDC25C, GADD45α, GADD45β, and GADD45γ, thereby inhibiting the function of CyclinB1 and CDK1 and affecting the progression of U87MG and GBM8401 cells through the G2/M phase of the cell cycle. Based on the Western blot analysis of G2/M-related proteins, it can be concluded that CDDO-dhTFEA has the ability to modulate the relative intensities of G2/M cell cycle proteins in U87MG and GBM8401 cells.

### 2.6. CDDO-dhTFEA Modulates Relative Intensities of Closely Associated G2/M Cell Cycle Genes in Glioblastoma Cells

To gain a comprehensive understanding of the effects of CDDO-dhTFEA on cellular pathways, we conducted an analysis of pathway enrichment using the Kyoto Encyclopedia of Genes and Genomes in U87MG and GBM8401 cells treating the synthetic triterpenoid. Through this analysis, we were able to identify the top enriched pathways in both cell lines. Among these pathways, the cell cycle pathway was found to be significantly affected by CDDO-dhTFEA treatment, indicating its potential role in activating signaling pathways related to the cell cycle. As shown Figure 7A, we performed a heatmap cluster analysis on 16 G2/M-related TPM genes (CCNB1, WEE1, CCNA1, CDK1, PCNA, CDC25C, CDC25B, CDC25A, Myt1, GADD45α, GADD45γ, GADD45β, TP53, CHEK1, CHEK2, and CDKN1A). The Z values of eight genes (CCNA1, Wee1, CDC25A, GADD45A, GADD45b, GADD45G, Myt1, and CDKN1A) were higher than those in the blank control group in GBM8401 cells. The Z values of 10 genes (CCNA1, CDK1, CDC25A, GADD45A, GADD45b, GADD45G, Myt1, CDKN1A, CHEK2, and CDC25C) were higher than those in the blank control group in U87MG cells. Treatment with CDDO-dhTFEA leads to changes in the mRNA expression levels of G2/M-related TPM genes. (Figure 7B,C). To assess changes in gene expression, we compared the expression levels between cells treated with CDDO-dhTFEA and cells treated with the vehicle control. We defined changes greater than 2-fold as significantly upregulated and changes less than 0.5-fold as downregulated. These results suggest that CDDO-dhTFEA treatment affects mRNA gene regulation related to the G2/M cell cycle.

## 3. Discussion

The proliferation of U87MG and GBM8401 cells was inhibited by CDDO-dhTFEA in a dose-dependent and time-dependent manner, as demonstrated by this study. Additionally, CDDO-dhTFEA regulates cell proliferation and significantly increases the DNA synthesis ratio. The research findings indicated that CDDO-dhTFEA treatment resulted in the inhibition of cell cycle progression at the G2/M phase, causing a delay in mitosis and potentially suppressing cell growth. By regulating protein and gene expression levels, treatment with CDDO-dhTFEA resulted in cell cycle G2/M arrest. Based on these findings, it can be concluded that CDDO-dhTFEA inhibits cell proliferation by inducing cell cycle G2/M arrest and regulating G2/M cell cycle gene expression.

CDDO and its derivatives have been shown to have pleiotropic effects [10]. At low doses, they activate Nrf2, which activates the Nrf2 signaling pathway [24], leading to the production of downstream antioxidant precursors [25]. Simultaneously, they inhibit the nuclear factor-kappa B signaling pathway [26], which inhibits the production of inflammatory precursors and results in antioxidant and anti-inflammatory effects. Therefore, CDDO and its derivatives have been used to treat chronic inflammatory diseases, such as Friedreich Ataxia [27] and chronic kidney disease [28]. Cytotoxic effects on tumor cells have been described for CDDO [29], derivatives that include CDDO-Me [30], CDDO-Im [20], CDDO-EA [20], CDDO-TFEA [22,23], and omaveloxolone [31] at moderate and high doses. As a result, CDDO and its derivatives could potentially exhibit anticancer activity at multiple levels. At low doses, they could block the carcinogenic process and reduce damage from carcinogens. At moderate to high doses, they could slow cancer cell proliferation or cause cell death through apoptosis. At low concentrations, CDDO and its derivatives have been found to activate the Nrf2/KEEP signaling pathway, which provides cytoprotective effects through antioxidant and anti-inflammatory mechanisms. However, at medium to high concentrations, these compounds have been shown to induce cytotoxicity through various signaling pathways. We found that CDDO-dhTFEA treatment reduced the cell viability in a dose-dependent manner.

Cell cycle checkpoints were first introduced by Hartwell [32] to guarantee the proper cell cycle progression. When DNA replication or chromosome segregation defects occur, cells undergo cell cycle arrest until the defects are repaired. There are three checkpoints in the cell cycle, including G1 [33], G2/M [34], and spindle [21]. Among them, the G2/M checkpoint is the most crucial in responding to genotoxic stimuli. It ensures that cells do not enter mitosis until DNA damage or incomplete replication is adequately repaired, thereby maintaining genomic stability [35]. In this study, CDDO-dhTFEA promotes DNA synthesis, causing cells to progress from the S phase to the G2 phase and CDDO-dhTFEA-induced mitotic arrest in the mitotic index analysis. After treatment with CDDO-dhTFEA, the cell cycle analysis revealed a notable rise in the amount of cells in the G2/M phase in both U87MG and GBM8401 cells. These findings are consistent with previous reports in the literature. For example, CDDO-imidazolide induces G2/M cell cycle arrest in BRCA1-mutated breast cancer cells [18], while CDDO-Me significantly arrests cells in the G2/M and S phases in K562 cells [19]. Additionally, observations indicate that treatment with CDDO-Me, CDDO-Im, and CDDO-TFEA resulted in a rise in sub-G1/G0 and significant reduction in S phase populations, while cells were arrested in the G2/M phase and could not proceed to the G1/G0 phase. [20,21]. Wang et al. [19] also reported that the expression levels of key cell cycle regulators were significantly altered when cells were arrested in the G2/M and S phases by CDDO-Me. Furthermore, Kim et al. [18] demonstrated that the ability of CDDO-Im to inhibit the BRCA1 mutant tumor G2/M cell cycle was associated with DNA damage followed by the activation of the DNA damage checkpoint. Finally, Tsai et al. showed that induction of G2/M arrest in the cell cycle may be attributed to CDDO-TFEA treatment in established GBM8401 [23,36] and U87MG cells [22,36]. Taken together, these findings provide evidence that CDDO derivatives such as CDDO-dhTFEA induce cell cycle arrest in the G2/M phase, likely through the regulation of G2/M cell cycle genes.

Uncontrolled cell cycle progression is a hallmark of cancer, and cyclin-dependent kinases (CDKs) and cyclins are key regulators of the normal cell cycle. CDK1 and cyclin B primarily regulate the G2/M transition. The cyclin B–CDK1 complex activity is regulated by CDK phosphorylation and dephosphorylation, which are inhibited by WEE1 and Myt1 phosphorylation of CDK1, respectively. Wee1 is a crucial regulator of the G2/M transition and inhibits CDK1 activity by phosphorylating CDK1 amino acids, which regulates G2/M transition, preventing mitosis [37]. Conversely, CDK1 activation is achieved through the dephosphorylation of CDK1 catalyzed by dephosphorylation enzymes, such as Cdc25c. Cdc25c-mediated dephosphorylation of CDK1 triggers mitosis. Cdc25 dephosphorylates Tyr15 to activate Cdk1 at the G2/M transition, while Wee1 is phosphorylated and inactivated. [38]. Additionally, CDK inhibitors such as p21 can control G2/M cell cycle progression by controlling CDK1 activity, and p21 inhibits the activity of CDK1-cyclin B, necessary for orderly progression through G2/M. Based on our observations, CDDO-dhTFEA treatment had distinct effects on different cell lines. On the one hand, treatment with CDDO-dhTFEA resulted in distinct relative intensities of protein patterns in U87MG and GBM8401 cells. Specifically, the relative expression levels of CDK1 and CyclinB1 were reduced, while in glioma cells, the expression levels of Wee1 and p21 were upregulated in GBM8401 cells. On the other hand, CDDO-dhTFEA treatment increased the relative intensities of CDK1 and Cyclin B proteins in U87MG cells. Therefore, the effect of CDDO-dhTFEA treatment on cellular mechanisms may vary depending on the type of cell line used.

The CDK1–cyclin complex and G2/M cell cycle regulation are modulated by various factors, including CDK1 phosphorylation kinase (Wee1), dephosphatase (Cdc25c), and inhibitor (p21). Our study, along with previous results in the literature, indicate that CDDO-dhTFEA affects these regulatory pathways and can influence the G2/M cell cycle (As Figure 8). However, it should be noted that the Western blot analysis showed different expression patterns of G2/M-related proteins in the two glioma cell lines after CDDO treatment, indicating a need for further investigation. Furthermore, the KEGG pathway enrichment analysis revealed that CDDO-dhTFEA treatment of U87MG and GBM8401 cells led to alterations in the same cell cycle pathway, suggesting a possible impact on signaling pathway activation. The heatmap cluster analysis showed that CDDO-dhTFEA treatment affected G2/M-related TPM gene regulation. To investigate the impact of 8 µM CDDO-dhTFEA on the gene expression profiles of G2/M phase-related genes, we performed next-generation sequencing analysis. These results revealed that CDDO-dhTFEA treatment caused significant changes in the expression levels of closely associate genes of the G2/M cycle in glioblasotma cells, supporting previous findings.

## 4. Materials and Methods

### 4.1. Reagents and Chemicals

Thermo Fisher/Invitrogen and Cayman Chemicals were the sources of the PrestoBlue™ Cell Viability Reagent and CDDO-dhTFEA, respectively. Sigma-Aldrich (St. Louis, MO, USA) provided the phosphate-buffered saline (PBS), RPMI 1640 medium, Trypan Blue solution, dimethyl sulfoxide (DMSO), and trypsin-EDTA (0.25%). Propidium iodide (PI) was purchased from Sigma-Aldrich, and markers were obtained from Bio-Rad Laboratories (Hercules, CA, USA). The following antibodies were used (all 1:1000 dilution): β-actin (Sigma-Aldrich, A5441), cyclin B and cyclin A (Proteintech, Chicago, IL, USA, 55004-1-AP), Bcl-2 (Sigma-Aldrich), CDK1 (Cell Signaling Technology, Danvers, MA, USA, E1Z6R), caspase-3 (Sigma-Aldrich), LC3B (Sigma-Aldrich), poly(ADP-ribose) polymerase (PARP, Sigma-Aldrich), beclin-1 (Sigma-Aldrich), Bax (Sigma-Aldrich), and p62/sequestosome 1 (SQSTM1, Sigma-Aldrich).

### 4.2. Cell Culture

The sources of the U87MG (astrocytoma) and GBM8401 (glioblastoma) were distinct. The U87MG cells were obtained from the Food Industry Research and Development Institute (Hsinchu, Taiwan) and cultured in MEM medium supplemented with 10% FBS and 1% P/S. In contrast, the GBM8401 cell line was sourced from the BCRC collects novel bioresources located in Hsinchu, Taiwan. All cells were incubated under standard conditions of 37 °C and 5% CO_2_ in RPMI medium supplemented with 1% penicillin-streptomycin (P/S) and 10% fetal bovine serum (FBS).

### 4.3. Cell Viability Assay

Using the 3-(4,5-dimethylthiazol-2-yl)-2,5-diphenyltetrazolium bromide (MTT) assay, cell viability was assessed according to a previously described protocol [39]. The culture plate was prepared for the U87MG and GBM8401 cells by seeding 3 × 10^3^ cells per well, followed by incubation under standard conditions of 37 °C, 5% CO_2_, and saturated humidity for 24 h. After 24 h, the cells were left untreated or treated with dimethyl sulfoxide (DMSO) or various concentrations (2, 4, and 8 µM) of CDDO-dhTFEA for 24–72 h. After the 24 h incubation period, each well containing U87MG and GBM8401 cells was treated with the PrestoBlue reagent and incubated for a minimum of 10 min. Using a multiwell plate reader, the absorbance was then measured. The IC50 values were calculated for all samples, and each sample was tested in triplicate.

### 4.4. Cell Cycle Analysis

In order to examine the impact of medication on the progression of the cell cycle, a cell cycle analysis was performed using methods described in a previous study [40]. Briefly, 3 × 10^5^ cells were seeded and incubated in culture dishes for 24 h, and then left untreated or exposed to DMSO or varying concentrations of CDDO-dhTFEA for 24 h. The cells were collected and preserved in 70% ethanol (3 mL), which was stored at −20 °C for at least 8 h following the incubation period. Subsequently, the ethanol was aspirated, and the cells were washed at least once with PBS. To measure the fluorescence of the cell DNA, the cells were treated with a solution containing propidium iodide, Triton X-100, and RNase A, which was then left to incubate for a period of 30 min. Then, a FACSCalibur flow cytometer (BD Biosciences, Santa Clara, CA, USA) was used to record the fluorescence signals. The data were analyzed using the freely available WinMDI 2.9 BD Biosciences software.

### 4.5. Cell Proliferation/DNA Synthesis Analysis

To evaluate DNA synthesis, 3 × 10^5^ cells were seeded and incubated in the culture dishes for 24 h, followed by being untreated or exposed to DMSO or various concentrations of CDDO-dhTFEA (2, 4, and 8 µM). Then, the cells were labeled with 5-ethynyl-2′-deoxyuridine (EdU) DNA for 24 h using the EZClick ™ EdU Cell Proliferation/DNA Synthesis Kit (FACS/Microscopy, BioVision, Waltham, MA, USA) [41]. Cells in the negative control group were not subjected to treatment with CDDO-dhTFEA, exposure to DNA labeling, or any other experimental manipulation, while cells in the positive control group were treated only with EdU DNA labeling. The cell DNA was stained with fluorescent azide and green using total DNA for 30 min. A FACSCalibur flow cytometer from BD Bioscience was utilized to detect and record the emitted green fluorescence, and all data were processed using the BD WinMDI 2.9 Biosciences software.

### 4.6. Mitotic Index Analysis

MPM-2 monoclonal antibody is widely used to assess mitotic disturbances. To assess the impact of drug treatment on cell division, we employed MPM-2 (anti-phospho-Ser/Thr-Pro) as a mitotic marker to evaluate the mitotic index of drug-treated cells [42]. Cells were initially placed in culture dishes (3 × 10^5^), incubated overnight, and left untreated or treated with CDDO-dhTFEA for 24 h. The U87MG and GBM8401 cells were subjected to treatment with nocodazole as a positive control, which is known to induce metaphase arrest, and served as the positive control in the experiment. The cells were treated with 70% ethanol and kept at −20 °C for a minimum of 8 h to ensure fixation, stained with 200 µL of IFA-Tx buffer containing 0.1% sodium azide, 0.1% Triton X-100, 150 nM/L NaCl, 10 nM/L 4-(2-hydroxyethyl)-1-piperazineethanesulfonic acid, and 4% fetal calf serum (FCS), along with an anti-mouse rabbit fluorescein isothiocyanate antibody (Serotec, Oxford, UK) and MPM-2 anti-phospho-Ser/Thr-Pro antibody for 1 h at room temperature in the dark. The cells were finally detected using FACSCalibur flow cytometry (BD Bioscience), and the functionality of the MPM-2 antibody was assessed in both the control and experimental groups using the freely available software, WinMDI 2.9 (BD Bioscience).

### 4.7. Western Blotting

Western blotting was conducted as described previously [43]. Lysis buffer was used to prepare cell lysates on ice, with a protein amount ranging from 50 to 80 µg. The protein samples were subjected to sodium dodecyl sulfate-polyacrylamide gel electrophoresis on 10% gels. After electrophoretic protein separation, the proteins were transferred to polyvinylidene fluoride (PVDF) membranes (Millipore, Billerica, MA, USA) for 2 h. Following an overnight blocking, the membranes were subjected to primary antibody incubation, including an antibody against β-actin, for either 2 h at room temperature or overnight at 4 °C, according to the methods described in the Materials and Methods section (Section 4. After primary antibody incubation and washing, the membranes were treated with a 1:20,000 dilution of the appropriate secondary antibody (LI-COR, Lincoln, NB, USA) for 30–40 min, followed by washing with PBST and PBS. The antigens were detected by either an Odyssey near-infrared fluorescence imaging system (LI-COR) or an enhanced chemiluminescence detection kit (Amersham Corp., Arlington Heights, IL, USA) after washing. Using the ImageJ software (NIH, Bethesda, MD, USA), normalization of documented values to β-actin was done after densitometry analysis, which included integrated density of bands.

### 4.8. Next Generation Sequencing (NGS)

Following a 24 h incubation period, RNA was extracted from the cells utilizing RNAzol RT reagent [44]. To start, one microgram of purified total RNA using oligo (dT) was employed. The eukaryotic mRNA was captured by magnetic beads, and heat was administered to break the mRNA into fragments. These fragments of mRNA served as a template for generating the first strand of cDNA using reverse transcriptase and random primers. Reagents such as DNA polymerase, RNase H, and dNTPs were added to purify cDNA using Beckman Coulter (Beverly, MA, USA). Subsequently, the cDNA underwent end repair and 3′ adenylation, followed by ligation of a sequencing adapter. After amplification and purification of the resulting products with PCR and magnetic beads, the library was sequenced. The size distribution of the fragments was determined from Agilent (San Diego, CA, USA), usinga BioAnalyzer2100 system. The concentration of the library was determined using a real-time PCR system. From Illumina (San Diego, CA, USA), size 150 PE sequencing was carried out using a NovaSeq 6000 sequencer.

### 4.9. Data Analysis

The statistical analysis was performed using the GraphPad software (GraphPad, San Diego, CA, USA) for one-way and two-way analyses of variance. The statistical significance was determined by setting the threshold for *p*-value at 0.05. The statistical analysis was performed using the SPSS 24.0 version software (SPSS Inc., Chicago, IL, USA). The data from at least three independent experiments are expressed as the mean ± SEM.

## 5. Conclusions

CDDO-dhTFEA, a synthetic derivative of OA, has emerged as a potential anticancer agent against glioblastoma cells. The proliferation of U87MG and GBM8401 cells was inhibited by CDDO-dhTFEA in a dose-dependent and time-dependent manner, as demonstrated by this study. Additionally, CDDO-dhTFEA regulates cell proliferation and significantly increases the DNA synthesis ratio. Its antiproliferative effects are the ability to induce cell cycle arrest in the G2/M phase, as well as modulation of the expression of various genes and proteins associated with this phase of the cell cycle in glioblastoma cells. Treatment with CDDO-dhTFEA led to cell cycle G2/M arrest and inhibited proliferation of U87MG and GBM8401 cells by regulating G2/M cell cycle proteins and gene expression in GBM cells in vitro. Although promising, the specific mechanisms underlying CDDO-dhTFEA’s effects and its complete range of functions require further investigation.

## Figures and Tables

**Figure 1 pharmaceuticals-16-00642-f001:**
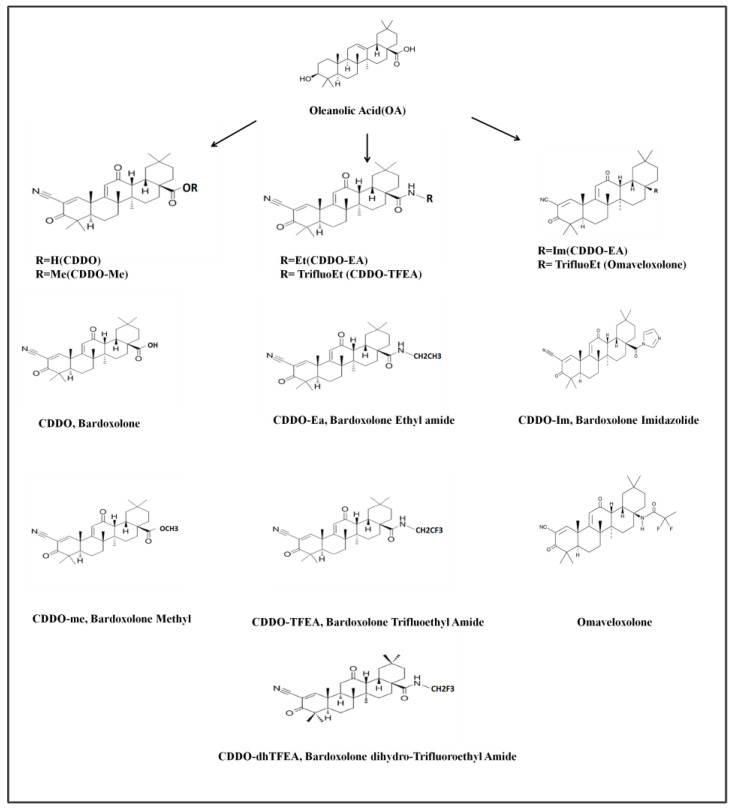
Structures of the synthetic oleanane triterpenoids.

**Figure 2 pharmaceuticals-16-00642-f002:**
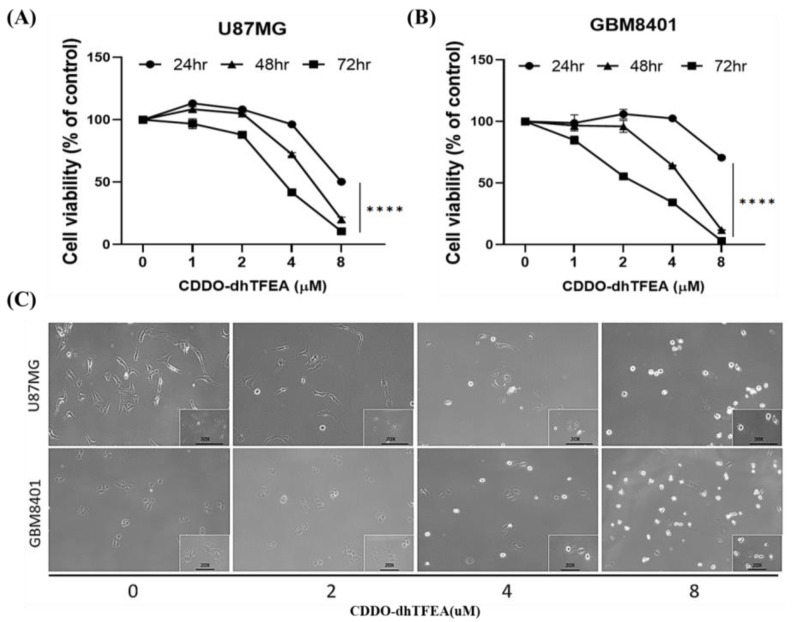
Using the PrestoBlue reagent assay, we evaluated the dose- and time-dependent effects of CDDO-dhTFEA on the cell viability in: (**A**) U87MG cells; (**B**) GBM8401 cells. The cells were incubated with various concentrations (ranging from 1 to 8 µM) of CDDO-dhTFEA or DMSO as the control for 24, 48, and 72 h. The results demonstrated that CDDO-dhTFEA reduced the cell viability in both cell types; (**C**) the morphologies of the cells exposed to CDDO-dhTFEA were different from those of the control cells, and the differences were concentration dependent. The data are presented as means ± SD of three independent experiments. Statistical analysis showed that the results were significant. **** *p* < 0.0001 compared with control.

**Figure 3 pharmaceuticals-16-00642-f003:**
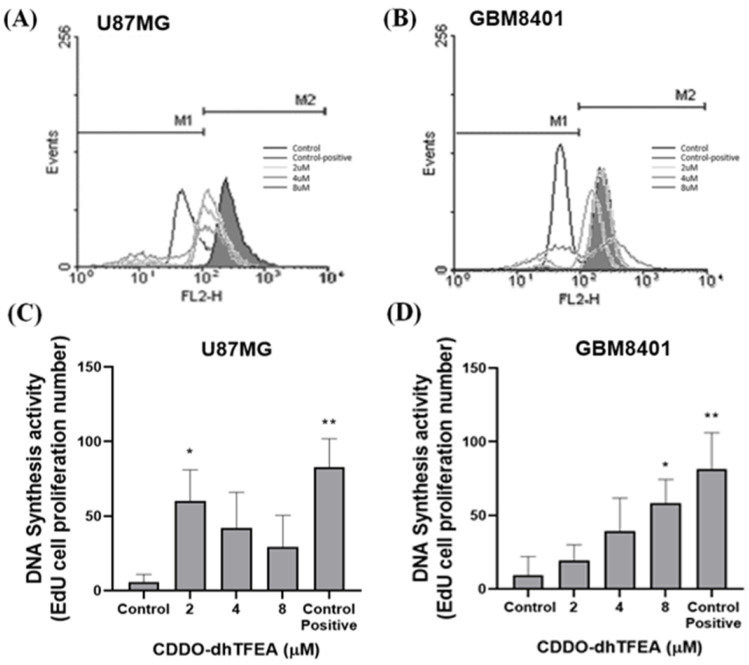
Regulation of DNA synthesis in glioma cells by CDDO-dhTFEA: (This experiment was to investigate the effect of CDDO-dhTFEA on DNA synthesis in glioblastoma cells such as U87MG and GBM8401 cells (**A**,**B**). EdU DNA labeling was used to assess DNA synthesis; The cells were then analyzed by flow cytometry. CDDO-dhTFEA significantly increased DNA synthesis in both U87MG and GBM8401 cells (**C**,**D**). The results of the study indicate that CDDO-dhTFEA could potentially enhance DNA synthesis in both U87MG and GBM8401 cells. * *p* < 0.05 and ** *p* < 0.01 compared with control.

**Figure 4 pharmaceuticals-16-00642-f004:**
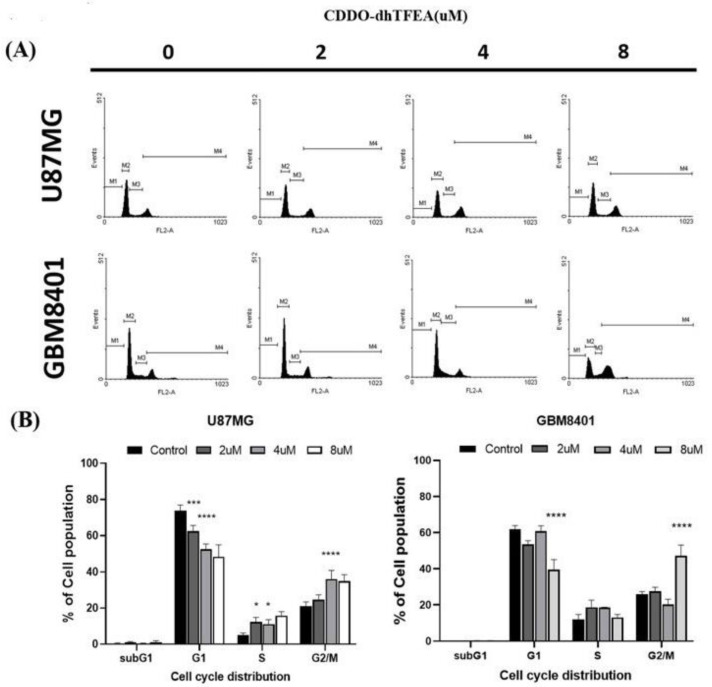
CDDO-dhTFEA triggers G2/M cell cycle progression accumulation in glioma cells: (**A**) Analysis of cell cycle in U87MG and GBM8401 cells treated with 0, 2, 4, and 8 µM CDDO-dhTFEA for 24 h; (**B**) using flow cytometry, induction of G2/M phase arrest by CDDO-dhTFEA. The cells were stained with propidium iodide and assessed. * *p* < 0.05, *** *p* < 0.001 and **** *p* < 0.0001 compared with control.

**Figure 5 pharmaceuticals-16-00642-f005:**
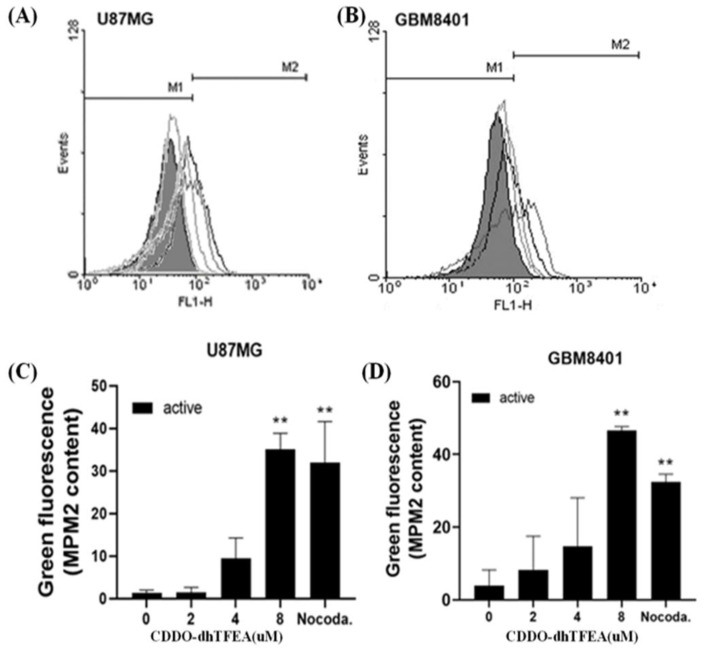
The treatment of glioma cells with CDDO-dhTFEA resulted in an increase in mitotic activity: (**A**,**B**) Flow cytometry analysis was performed to evaluate the effects of CDDO-dhTFEA treatment on glioma cells. Briefly, glioma cells were incubated with CDDO-dhTFEA at various concentrations (from 0 to 8 µM) for 24 h, and following treatment, the cells were fixed with 70% ethanol, stained with MPM-2 and PI, and subjected to flow cytometry analysis; (**C**,**D**) quantification of MPM2 content showed that the expression of MPM2 was found to be increased in glioblastoma cells upon treatment with CDDO-dhTFEA. Nocodazole (10 µg/mL), an antifungal agent that induces metaphase arrest, was used as the positive control. ** *p* < 0.01 compared with control.

**Figure 6 pharmaceuticals-16-00642-f006:**
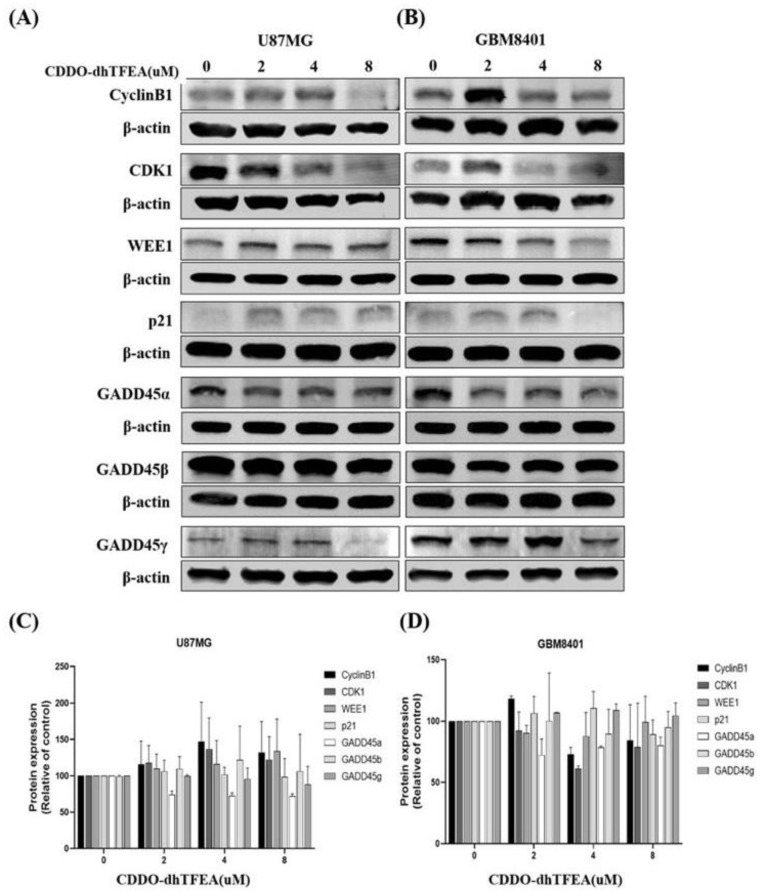
(**A**,**B**) The expression levels of G2/M closely associated proteins in glioblastoma cells treated with CDDO-dhTFEA were assessed by using a Western blot analysis. (**C**,**D**)The cells were treated with CDDO-dhTFEA at various concentrations (ranging from 0 to 8 µM) for a duration of 24 h, and resulted in a significant reduction in the expression levels of CyclinB1, CDK1, GADD45α, GADD45β, and GADD45γ in the GBM8401 cells. In addition, both cell lines exhibited upregulation of Wee1 and CDC25C expression. Furthermore, it resulted in an increase in the relative intensities of p21. These proteins play a critical role in regulating the G2/M cell cycle.

**Figure 7 pharmaceuticals-16-00642-f007:**
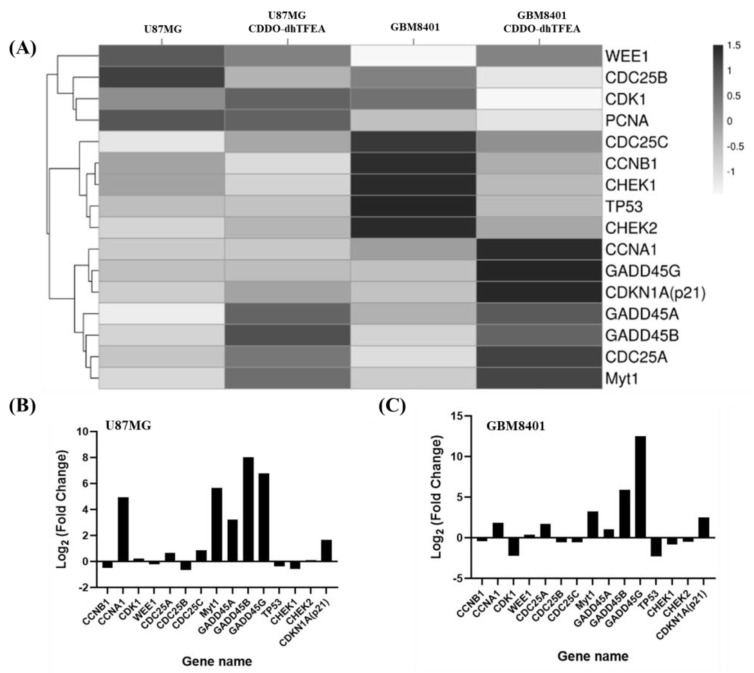
(**A**) Heatmap clustering analysis was conducted to assess the expressed genes following treatment with CDDO-dhTFEA compared to the control; (**B**,**C**) to confirm the mRNA results of CDDO-dhTFEA treatment, an NGS analysis was conducted on glioma cells. The analysis identified genes that were either upregulated or downregulated in response to CDDO-dhTFEA treatment. Specifically, the expression profiles of genes related to the G2/M phase, as measured by the log fold change (log Fc), were investigated in cells exposed to 8 µM CDDO-dhTFEA for 24 h.

**Figure 8 pharmaceuticals-16-00642-f008:**
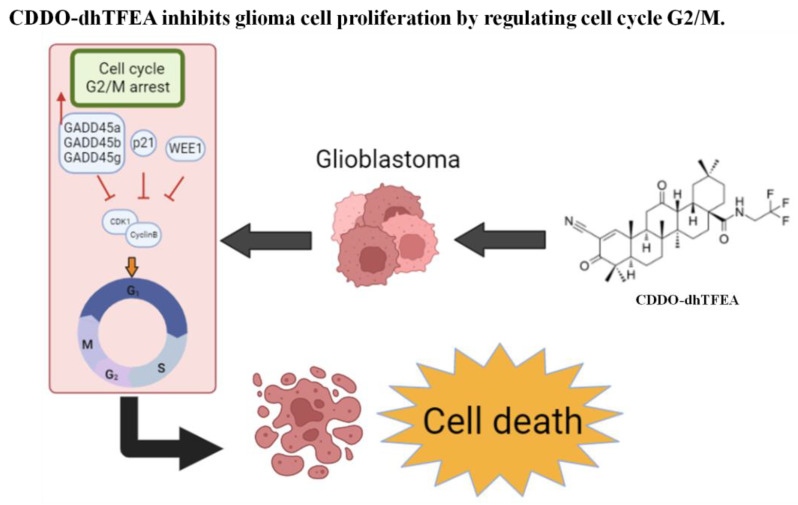
CDDO-dhTFEA inhibits glioma cell proliferation by regulating cell cycle G2/M.

## Data Availability

All the data produced or scrutinized during the course of this research have been incorporated in this article.

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
