# Peer review of "A Novel Synthetic Oleanolic Acid Derivative Inhibits Glioma Cell Proliferation by Regulating Cell Cycle G2/M Arrest"

_pharmaceuticals, 2023, doi:10.3390/ph16050642_

Round 1

Reviewer 1 Report

The submitted manuscript described the action of the compound CDDO-dhTFEA, a synthetic derivate of oleanolic acid. The effect of the compound was tested in GBM8401 and U87MG cells using viability assay, cell cycle analysis and mRNA and protein expression of CC regulating genes.

Methodology and results are in general clearly presented. What is the concentration of DMSO at the highest concentration tested? What are the “positive controls”? Where are the data on apoptosis since PARP, Caspase, Bax, etc were tested according to Materials and Methods.

Major shortcomings are that novelty of the findings is not clear, only the effect on CC was studied in detail, no comparison with other derivates of oleanolic acid are presented to compare efficacy with existing compounds, and that action on normal cells was not studied. Important questions remain unanswered:

Do other derivates of oleanolic acid act by similar mechanisms?

What is their effect on normal cells?

Minor

Few spelling errors should be corrected like in l.319, l.264

Author Response

Dear Reviewer 1:

Thank you for giving me the opportunity to submit a revised draft of our manuscript which titled: A novel synthetic oleanolic acid derivative inhibits glioma cell proliferation by regulating cell cycle G2/M arres. We appreciate the time and effort that you and the reviewers have dedicated to providing your valuable feedback on our manuscript. We are grateful to the reviewers for their insightful comments on this paper. We have been able to incorporate changes to reflect most of the suggestions provided by the reviewers. We have highlighted the modification within the manuscript and point-by-point response to the reviewers’ comments and concerns in this revision.

Comments from Reviewer 1

The submitted manuscript described the action of the compound CDDO-dhTFEA, a synthetic derivate of oleanolic acid. The effect of the compound was tested in GBM8401 and U87MG cells using viability assay, cell cycle analysis and mRNA and protein expression of CC regulating genes.

Comment 1: Methodology and results are in general clearly presented. What is the concentration of DMSO at the highest concentration tested?

Response to the comment 1:

Thank you for pointing this out. We thank the reviewer for bringing up this question. The cell viability assay was performed by treating the cells with dimethyl sulfoxide (DMSO) or various concentrations (2, 4, and 8 µM) of CDDO-dhTFEA for 24-72 hours. The concentration of DMSO used will be less than 0.1% regardless of the various concentrations of CDDO-dhTFEA (2, 4, and 8 µM). Therefore, when the highest concentration of CDDO-dhTFEA used is 8 µM, the concentration of DMSO used to process GBM8401 and U87MG cells will also be 0.1% lower. (Line347 -Line349; Line367 -Line368)

Comment 2: Methodology and results are in general clearly presented. What are the “positive controls”?

Response to the comment 2:

Thank you for pointing this out. There are many kinds of Nrf2 activator, CDDO and derivatives such as CDDO-MA, CDDO-Im, CDDO-TFEA and CDDO–dhTFEA in our laboratory. We have accordingly modified the manuscript to clarify the rationale for selecting CDDO-TFEA as the positive control in our experiments (As below Figure; data not shown). We chose this drug based on our previous research (As References) and the fact that CDDO-dhTFEA is synthesized from CDDO-TFEA.

In addition, we used Nocodazole as a positive control in our cell proliferation/DNA synthesis experiment. Nocodazole is a potent inhibitor of tubulin production and mitosis, and induces cancer cells to arrest in the M phase. By adding Nocodazole as a control group, we were able to compare the effect of the experimental drug on the cell cycle of cancer cells. Thank you to the reviewer for bringing up this point. (Line367 -Line368)

References:

  1. Tsai, T.-H.; Su, Y.-F.; Tsai, C.-Y.; Wu, C.-H.; Lee, K.-T.; Hsu, Y.-C. RTA dh404 Induces Cell Cycle Arrest, Apoptosis, and Autophagy in Glioblastoma Cells.  J. Mol. Sci.202324, 4006. https://doi.org/10.3390/ijms24044006
  2. Tsai TH, Lieu AS, Huang TY, Kwan AL, Lin CL, Hsu YC. Induction of Mitosis Delay and Apoptosis by CDDO-TFEA in Glioblastoma Multiforme. Front Pharmacol. 2021 Nov 8;12:756228. doi: 10.3389/fphar.2021.756228. PMID: 34858180; PMCID: PMC8630575.

Comment 3: Where are the data on apoptosis since PARP, Caspase, Bax, etc were tested according to Materials and Methods.

Response to the comment 3:

Thank you for pointing this out. We have accordingly modified the manuscript and results to emphasize this point. We would like to thank the reviewer for bringing up this question. In this experiment, our focus was on investigating the anti-cancer effects of CDDO-dhTFEA. While it is true that CDDO-dhTFEA can induce apoptosis and autophagy, what is even more significant is its impact on the cell cycle. The subsequent genes and proteins expression have confirmed the important anti-cancer effect of CDDO-dhTFEA due to the influence on the cell cycle. Therefore, the experimental design was mainly focused on genes and proteins related to the cell cycle. Even specific in G2/M cell cycle related genes and proteins. We agree with the reviewer's suggestion that the apoptosis-related proteins PARP, Caspase, and Bax are important research directions, and we believe they are worthy of further design and implementation of follow-up research.

Comment 4: Major shortcomings are that novelty of the findings is not clear, only the effect on CC was studied in detail, no comparison with other derivates of oleanolic acid are presented to compare efficacy with existing compounds, and that action on normal cells was not studied.

Response to the comment 4:

Thank you for pointing this out. We agree with this comment. There are many kinds of Nrf2 activator, CDDO and derivatives such as CDDO-MA, CDDO-Im, CDDO-TFEA and CDDO–dhTFEA in our laboratory. CDDO and its derivatives are synthetic oleanane triterpenoid compounds that potently activate Nrf2 and inhibit the pro-inflammatory transcription factor, NF-κB. CDDO and its derivatives increased Nrf2 expression, decreased oxidative stress, decreased NF-κB, and decreased the levels of proinflammatory mediators. Therefore, CDDO and its derivatives work through a similar mechanism. We have accordingly modified the manuscript to clarify the rationale for selecting CDDO-TFEA as the positive control in our experiments (As below Figure; data not shown). We chose this drug based on our previous research (As References) and the fact that CDDO-dhTFEA is synthesized from CDDO-TFEA.  

In addition, we had been also detects in normal lung fibroblasts MRC-5 cells. There was no change in normal lung fibroblasts MRC-5 cells (data not shown). In addition, adding drugs to treat normal brain cells to calculate a selective index is a very good suggestion, but the current culture technology cannot cultivate normal brain cells. Therefore, we use normal lung fibroblasts MRC-5 cells as a control group for proliferation. There was no change in normal lung fibroblasts MRC-5 cells. (Line98 –Line99)

References:

  1. Tsai, T.-H.; Su, Y.-F.; Tsai, C.-Y.; Wu, C.-H.; Lee, K.-T.; Hsu, Y.-C. RTA dh404 Induces Cell Cycle Arrest, Apoptosis, and Autophagy in Glioblastoma Cells.  J. Mol. Sci.202324, 4006. https://doi.org/10.3390/ijms24044006
  2. Tsai TH, Lieu AS, Huang TY, Kwan AL, Lin CL, Hsu YC. Induction of Mitosis Delay and Apoptosis by CDDO-TFEA in Glioblastoma Multiforme. Front Pharmacol. 2021 Nov 8;12:756228. doi: 10.3389/fphar.2021.756228. PMID: 34858180; PMCID: PMC8630575.

Comment 5: Important questions remain unanswered: Do other derivates of oleanolic acid act by similar mechanisms? What is their effect on normal cells?

Response to the comment 5:

Thank you for pointing this out. We agree with this comment. There are many kinds of Nrf2 activator, CDDO and derivatives such as CDDO-MA, CDDO-Im, CDDO-TFEA and CDDO–dhTFEA in our laboratory. CDDO and its derivatives are synthetic oleanane triterpenoid compounds that potently activate Nrf2 and inhibit the pro-inflammatory transcription factor, NF-κB. CDDO and its derivatives increased Nrf2 expression, decreased oxidative stress, decreased NF-κB, and decreased the levels of proinflammatory mediators. Therefore, CDDO and its derivatives work through a similar mechanism. We have accordingly modified the manuscript to clarify the rationale for selecting CDDO-TFEA as the positive control in our experiments. We chose this drug based on previous research and the fact that CDDO-dhTFEA is synthesized from CDDO-TFEA.

In addition, we had been also detects in normal lung fibroblasts MRC-5 cells. There was no change in normal lung fibroblasts MRC-5 cells (data not shown). In addition, adding drugs to treat normal brain cells to calculate a selective index is a very good suggestion, but the current culture technology cannot cultivate normal brain cells. Therefore, we use normal lung fibroblasts MRC-5 cells as a control group for proliferation. There was no change in normal lung fibroblasts MRC-5 cells. (Line98 –Line99)

Comment 6: Few spelling errors should be corrected like in l.319, l.264

Response to the comment 6:

Thank you for pointing this out. Thank you for this suggestion. We have accordingly modified the manuscript and results to emphasize this point. We have modified the manuscript to emphasize this point. Line 319 「Casase-3à Caspase-3」Line 264「Wang et al. [19] also reported that The…à Wang et al. [19] also reported that the」 In fact, the author has edited this document by the language teacher before submitting it, and we also ask the teacher to edit it again after the article is revised and submit it again. We try to minimize grammatical structural errors and typos as possible and typical English was used in this article. (Line330 –Line331; Line273 –Line274)

Additional clarifications

In addition to the above comments, all spelling and grammatical errors pointed out by the reviewers have been corrected.

Sincerely,

Prof. Tai-Hsin Tsai

Department of Neurosurgery, Kaohsiung Medical University Hospital

No. 100, Tzyou 1st Road, Sham-min District, Kaohsiung City, Taiwan

Reviewer 2 Report

The manuscript entitled “A novel synthetic oleanolic acid derivative inhibits glioma cell proliferation by regulating cell cycle G2/M arrest” aimed to investigate the ability of 2-Cyano-3,12-dioxooleana-1,9(11)-dien-28-oic acid-9,11-dihydro-trifluoroethyl amide to kill cancer in glioblastoma cells. This work is interesting and can provide a new methodology to fight cancer which is a global healthcare problem and the whole world needs new methods to treat this disease. However, I have the following comments:

1.      The authors have to mention in the Abstract, the used positive control which used in their study to validate their method.

2.      In the Introduction kindly write a short paragraph about the ability of oleanolic acid in crossing brain barrier or not.

3.      The conclusion part is very short and needs many works.

4.      For each used biological assay add suitable references.

5.      The whole manuscript needs major grammar, typo, and editing corrections by a native speaker

Author Response

Dear Reviewer 2:

Thank you for giving me the opportunity to submit a revised draft of our manuscript which titled: A novel synthetic oleanolic acid derivative inhibits glioma cell proliferation by regulating cell cycle G2/M arres. We appreciate the time and effort that you and the reviewers have dedicated to providing your valuable feedback on our manuscript. We are grateful to the reviewers for their insightful comments on this paper. We have been able to incorporate changes to reflect most of the suggestions provided by the reviewers. We have highlighted the modification within the manuscript and point-by-point response to the reviewers’ comments and concerns in this revision.

Comments from Reviewer 2:

The manuscript entitled “A novel synthetic oleanolic acid derivative inhibits glioma cell proliferation by regulating cell cycle G2/M arrest” aimed to investigate the ability of 2-Cyano-3,12-dioxooleana-1,9(11)-dien-28-oic acid-9,11-dihydro-trifluoroethyl amide to kill cancer in glioblastoma cells. This work is interesting and can provide a new methodology to fight cancer which is a global healthcare problem and the whole world needs new methods to treat this disease. However, I have the following comments:

Comment 1:  The authors have to mention in the Abstract, the used positive control which used in their study to validate their method.

Response to the comment 1:

Thank you for pointing this out. There are many kinds of Nrf2 activator, CDDO and derivatives such as CDDO-MA, CDDO-Im, CDDO-TFEA and CDDO–dhTFEA in our laboratory. We have accordingly modified the manuscript to clarify the rationale for selecting CDDO-TFEA as the positive control in our experiments (As below Figure; data not shown). We chose this drug based on our previous research (As References) and the fact that CDDO-dhTFEA is synthesized from CDDO-TFEA.

In addition, we used Nocodazole as a positive control in our cell proliferation/DNA synthesis experiment. Nocodazole is a potent inhibitor of tubulin production and mitosis, and induces cancer cells to arrest in the M phase. By adding Nocodazole as a control group, we were able to compare the effect of the experimental drug on the cell cycle of cancer cells. Thank you to the reviewer for bringing up this point. (Line367 -Line368)

References:

  1. Tsai, T.-H.; Su, Y.-F.; Tsai, C.-Y.; Wu, C.-H.; Lee, K.-T.; Hsu, Y.-C. RTA dh404 Induces Cell Cycle Arrest, Apoptosis, and Autophagy in Glioblastoma Cells.  J. Mol. Sci.202324, 4006. https://doi.org/10.3390/ijms24044006
  2. Tsai TH, Lieu AS, Huang TY, Kwan AL, Lin CL, Hsu YC. Induction of Mitosis Delay and Apoptosis by CDDO-TFEA in Glioblastoma Multiforme. Front Pharmacol. 2021 Nov 8;12:756228. doi: 10.3389/fphar.2021.756228. PMID: 34858180; PMCID: PMC8630575.

Comment 2: In the Introduction kindly write a short paragraph about the ability of oleanolic acid in crossing brain barrier or not.

Response to the comment 2:

Agree. We have accordingly modified the manuscript and results to emphasize this point. We have written a short paragraph in the Introduction about whether oleanolic acid has the ability to cross the brain barrier. 「However, one of the challenges of using oleanolic acid or its derivatives for the treatment of brain tumors is the ability of the compound to cross the blood-brain barrier (BBB). Studies have shown that oleanolic acid has low bioavailability and poor BBB penetration due to its high molecular weight and low lipophilicity[1,2]. However, recent research has explored different strategies to improve the BBB permeability of oleanolic acid, including the use of nanoparticles, drug delivery systems, and chemical modifications[3].These approaches may offer new opportunities for the development of oleanolic acid-based therapies for brain diseases.」(Line45Line52)

References

1.Castellano, J.M.; Ramos-Romero, S.; Perona, J.S. Oleanolic Acid: Extraction, Characterization and Biological Activity. Nutrients 2022, 14, doi:10.3390/nu14030623.

2.Pollier, J.; Goossens, A. Oleanolic acid. Phytochemistry 2012, 77, 10-15, doi:10.1016/j.phytochem.2011.12.022.

3.Ayeleso, T.B.; Matumba, M.G.; Mukwevho, E. Oleanolic Acid and Its Derivatives: Biological Activities and Therapeutic Potential in Chronic Diseases. Molecules (Basel, Switzerland) 2017, 22, doi:10.3390/molecules22111915.

Comment 3: The conclusion part is very short and needs many works.

Response to the comment 3:

Thank you for pointing this out. We agree with this comment. We have accordingly modified the manuscript and results to emphasize this point. The conclusion part is really short, so a lot of specific conclusions are put forward for the experimental results.「The proliferation of U87MG and GBM8401 cells was inhibited by CDDO-dhTFEA in a dose-dependent and time-dependent manner, as demonstrated by this study. Additionally, CDDO-dhTFEA regulates cell proliferation and significantly increases the DNA synthesis ratio….Treatment with CDDO-dhTFEA led to cell cycle G2/M arrest and inhibited proliferation of U87MG and GBM8401 cells by regulating G2M cell cycle proteins and gene expression in GBM cells in vitro.」(Line434 –Line436; Line440 –Line441)

Comment 4:  For each used biological assay add suitable references.

Response to the comment 4:

Thank you for this suggestion. Based on the reviewer's suggestion, we have added appropriate references for all biological assays used:

  1. Cell viability assay: Mosmann, T. Rapid colorimetric assay for cellular growth and survival: application to proliferation and cytotoxicity assays. Journal of immunological methods 1983, 65, 55-63, doi:10.1016/0022-1759(83)90303-4. (Line343 –Line343)
  2. Cell cycle analysis: Riccardi, C.; Nicoletti, I. Analysis of apoptosis by propidium iodide staining and flow cytometry. Nature protocols 2006, 1, 1458-1461, doi:10.1038/nprot.2006.238. (Line353 –Line354)
  3. Cell proliferation/DNA synthesis analysis: Flomerfelt FA, Gress RE. Analysis of Cell Proliferation and Homeostasis Using EdU Labeling. Methods Mol Biol. 2016;1323:211-20. doi: 10.1007/978-1-4939-2809-5_18. PMID: 26294411; PMCID: PMC7490834. (Line368 –Line369)
  4. Mitotic index analysis: Goto, H.; Tomono, Y.; Ajiro, K.; Kosako, H.; Fujita, M.; Sakurai, M.; Okawa, K.; Iwamatsu, A.; Okigaki, T.; Takahashi, T.; et al. Identification of a novel phosphorylation site on histone H3 coupled with mitotic chromosome condensation. The Journal of biological chemistry 1999, 274, 25543-25549, doi:10.1074/jbc.274.36.25543. (Line378 –Line381)
  5. Western blotting: Zheng, W.H.; Quirion, R. Glutamate acting on N-methyl-D-aspartate receptors attenuates insulin-like growth factor-1 receptor tyrosine phosphorylation and its survival signaling properties in rat hippocampal neurons. The Journal of biological chemistry 2009, 284, 855-861, doi:10.1074/jbc.M807914200. (Line394 –Line395)
  6. Next generation sequencing (NGS): Hong J, Gresham D. Incorporation of unique molecular identifiers in TruSeq adapters improves the accuracy of quantitative sequencing. Biotechniques. 2017 Nov 1;63(5):221-226. doi: 10.2144/000114608. PMID: 29185922; PMCID: PMC7359820. (Line412 –Line413)

Comment 5:  The whole manuscript needs major grammar, typo, and editing corrections by a native speaker

Response to the comment 5:

Agree. Thank you for this suggestion. It would have been interesting to explore this aspect. Thanks to the reviewer for pointing out this question. Agree. We have modified the manuscript to emphasize this point. In fact, the author has edited this document by the language teacher before submitting it, and we also ask the teacher to edit it again after the article is revised and submit it again. We try to minimize grammatical structural errors and typos as possible and typical English was used in this article.

Additional clarifications

In addition to the above comments, all spelling and grammatical errors pointed out by the reviewers have been corrected.

Sincerely,

Prof. Tai-Hsin Tsai

Department of Neurosurgery, Kaohsiung Medical University Hospital

No. 100, Tzyou 1st Road, Sham-min District, Kaohsiung City, Taiwan

Reviewer 3 Report

Reviewer comments and suggestions

The author of this study investigated the possibility of 

2-Cyano-3,12-dioxooleana-1,9(11)-dien-28-oic acid-9,11-dihydro-trifluoroethyl amide 

(CDDO-dhTFEA) as a potential cancer-fighting treatment in glioblastoma cells. The cell lines used were U87MG and GBM8401cells, and the result found that CDDO-dhTFEA was effective in reducing cell proliferation in both cell lines.

The authors also observed that CDDO-dhTFEA induced G2/M cell cycle arrest and mitotic delay, which may be associated with the inhibition of proliferation. The study concluded that CDDO-dhTFEA inhibits proliferation of U87MG and GBM8401cells by inducing cell cycle G2/M arrest and regulating G2M cell cycle proteins and gene expression in GBM cells in vitro. The paper was nicely written and could be accepted after following changes from my side. 

Below are the comments. 

  1. Please simply modify the sentence, it seems that both sentences could be merge “Treatment with CDDO-dhTFEA led to cell 27 cycle G2/M arrest associated with the regulation of protein and gene expression. CDDO-dhTFEA 28 inhibits proliferation of U87MG and GBM8401cells by inducing cell cycle G2/M arrest and regu- 29 lating G2M cell cycle proteins and gene expression in GBM cells in vitro”
  2. Line 47 needs to add the full form here as well, typo error in line 54.
  3. Lines 58-59 need more references here.
  4. Line 69071 No need to highlight the result here, so better if you can delete the line from the introduction.
  5. Line 107-108 What could be the explanation for this.
  6. Line 120 Please mention the full form of the x-axis (RTA dh) at the legend part
  7. Section 2.6 heat map It need to explore more.
  8. Line 227 The sentence was repeated.
  9. Line 290 a typo error 
  10. Please use the original Figure 1, as the provided picture was not clear, same comment for figure 3
  11. All references need to be modified by based on the MDPI guidelines. 

Author Response

Dear Reviewer 3:

Thank you for giving me the opportunity to submit a revised draft of our manuscript which titled: A novel synthetic oleanolic acid derivative inhibits glioma cell proliferation by regulating cell cycle G2/M arres. We appreciate the time and effort that you and the reviewers have dedicated to providing your valuable feedback on our manuscript. We are grateful to the reviewers for their insightful comments on this paper. We have been able to incorporate changes to reflect most of the suggestions provided by the reviewers. We have highlighted the modification within the manuscript and point-by-point response to the reviewers’ comments and concerns in this revision.

Comments from Reviewer 3:

The author of this study investigated the possibility of 2-Cyano-3,12-dioxooleana-1,9(11)-dien-28-oic acid-9,11-dihydro-trifluoroethyl amide  (CDDO-dhTFEA) as a potential cancer-fighting treatment in glioblastoma cells. The cell lines used were U87MG and GBM8401cells, and the result found that CDDO-dhTFEA was effective in reducing cell proliferation in both cell lines.The authors also observed that CDDO-dhTFEA induced G2/M cell cycle arrest and mitotic delay, which may be associated with the inhibition of proliferation. The study concluded that CDDO-dhTFEA inhibits proliferation of U87MG and GBM8401cells by inducing cell cycle G2/M arrest and regulating G2M cell cycle proteins and gene expression in GBM cells in vitro. The paper was nicely written and could be accepted after following changes from my side. 

Below are the comments. 

Comment 1: Please simply modify the sentence, it seems that both sentences could be merge “Treatment with CDDO-dhTFEA led to cell cycle G2/M arrest associated with the regulation of protein and gene expression. CDDO-dhTFEA inhibits proliferation of U87MG and GBM8401cells by inducing cell cycle G2/M arrest and regulating G2M cell cycle proteins and gene expression in GBM cells in vitro”

Response to the comment 1:

Thank you for pointing this out. We have accordingly modified the manuscript and results to emphasize this point. We will the following two sentences 「1.Treatment with CDDO-dhTFEA led to cell cycle G2/M arrest associated with the regulation of protein and gene expression.2. proliferation of U87MG and GBM8401cells by inducing cell cycle G2/M arrest and regulating G2M cell cycle proteins and gene expression in GBM cells in vitro」 has been merged 「Treatment with CDDO-dhTFEA led to cell cycle G2/M arrest and inhibited proliferation of U87MG and GBM8401 cells by regulating G2M cell cycle proteins and gene expression in GBM cells in vitro.(Line29–Line31)

Comment 2: Line 47 needs to add the full form here as well, typo error in line 54.

Response to the comment 2:

Thanks to the reviewer for pointing out this question. Agree. We have modified the manuscript to emphasize this point. We have changed lines 47 from「 CDDO to 2-cyano-3,12-dioxoolean-1,9-dien-28-oic acid (CDDO)」. We try to minimize grammatical structural errors and typos as possible and typical English was used in this article. (Line56–Line57: Line62 –Line63)

Comment 3: Lines 58-59 need more references here.

Response to the comment 3:

Thank you for pointing this out. We agree with this comment. We have added more references on lines 58-59. (Line68 –Line70)

References:

  1. Xie C, Zhou X, Liang C, Li X, Ge M, Chen Y, Yin J, Zhu J, Zhong C. Apatinib triggers autophagic and apoptotic cell death via VEGFR2/STAT3/PD-L1 and ROS/Nrf2/p62 signaling in lung cancer. J Exp Clin Cancer Res. 2021 Aug 24;40(1):266. doi: 10.1186/s13046-021-02069-4. Erratum in: J Exp Clin Cancer Res. 2021 Nov 6;40(1):349. PMID: 34429133; PMCID: PMC8385858.
  2. Saleem M, Asif J, Asif M, Saleem U. Amygdalin from Apricot Kernels Induces Apoptosis and Causes Cell Cycle Arrest in Cancer Cells: An Updated Review. Anticancer Agents Med Chem. 2018;18(12):1650-1655. doi: 10.2174/1871520618666180105161136. PMID: 29308747.

Comment 4: Line 69-71 No need to highlight the result here, so better if you can delete the line from the introduction.

Response to the comment 4:

Thanks to the reviewer for pointing out this question. Agree. We do not need to highlight the results of our experiments in the introduction so we will delete "The results from our experiments suggest that CDDO-dhTFEA treatment led to a significant suppression of cell proliferation in both U87MG and GBM8401cells and coincided with an increase in G2/M cell cycle arrest.” (Line79 –Line80)

Comment 5: Line 107-108 What could be the explanation for this.

Response to the comment 5:

Thank you for your suggestion. We agree with your comment that DNA synthesis increased linearly in GBM8401 cells and decreased linearly in U87MG cells with increasing drug concentration. 「Since the malignant degree of these two cell lines is different (GBM8401 is Grade 4 glioma (GBM) and U87MG is Mailganat glioma Grade 3), we observed different results in the EdU assay. The cell cycle arrest was more apparent in GBM8401, and we obtained similar results in gene and protein expression.」 However, we acknowledge that the differences between these two cell lines are important and may impact the interpretation of our results. We appreciate the reviewer's feedback, and we will consider these differences in the design of future studies to further explore the effects of these drugs on different cell lines. (Line119 –Line122)

Comment 6: Line 120 Please mention the full form of the x-axis (RTA dh) at the legend part.

Response to the comment 6:

Thank you for this suggestion. We have accordingly modified the manuscript and results to emphasize this point. Changed full form of all legend sections mentioning x-axis (RTA dh) to CDDO-dhTFEA. We have been rephrased and quoted the result text to provide for all data presented in this manuscript. (As all Figures)

Comment 7: Section 2.6 heat map It need to explore more.

Response to the comment 7:

Thanks to the reviewers for pointing out this problem. We have accordingly modified the manuscript and results to emphasize this point. We would like to thank the reviewer for bringing up this question. In this experiment, our focus was on investigating the anti-cancer effects of CDDO-dhTFEA. While it is true that CDDO-dhTFEA can induce apoptosis and autophagy, what is even more significant is its impact on the cell cycle. The subsequent gene and protein expression have confirmed the important anti-cancer effect of CDDO-dhTFEA due to the influence on the cell cycle. Even specific in G2/M cell cycle related genes and proteins. Therefore, the experimental design was mainly focused on genes and proteins related to the cell cycle. In the heat map of Section 2.6, we have shown as much as possible the genes and mRNAs related to the G2/M cell cycle (Line197 –Line218)

Comment 8: Line 227 The sentence was repeated.

Response to the comment 8:

Thanks to the reviewers for pointing out this problem; and we have been deleted the repeated sentence「At low doses, they possess anti-inflammatory and antioxidative properties, while at moderate doses, they induce cell differentiation. At high doses, they exhibit cytotoxicity. CDDO and its derivatives have dual effects on cell protection and cytotoxicity.」(Line239 –Line240)

Comment 9: Line 290 a typo error.

Response to the comment 9:

Thank you for this suggestion. We have accordingly modified the manuscript and results to emphasize this point. We have been rephrased the Line 290 a typo error.「whilegliomaàwhile glioma」 (Line299 –Line300)

Comment 10: Please use the original Figure 1, as the provided picture was not clear, same comment for figure 3.

Response to the comment 10:

Thank you for this suggestion. We have accordingly modified the Figures and results to emphasize this point.  (As Figure 2 and Figure 4)

Comment 11: All references need to be modified by based ohe MDPI guidelines.

Response to the comment 11:

Thank you for this suggestion. We agree with this comment. We have modified all references according to MDPI guidelines using Endnote software. (As All References )

Additional clarifications

In addition to the above comments, all spelling and grammatical errors pointed out by the reviewers have been corrected.

Sincerely,

Prof. Tai-Hsin Tsai

Department of Neurosurgery, Kaohsiung Medical University Hospital

No. 100, Tzyou 1st Road, Sham-min District, Kaohsiung City, Taiwan

Round 2

Reviewer 1 Report

I have no further comments.

Reviewer 2 Report

The authors conducted all the required corrections